# ESI mutagenesis: a one-step method for introducing mutations into bacterial artificial chromosomes

Arnaud Rondelet[1,*], Andrei Pozniakovsky[2,*], Devika Namboodiri[1] ✪, Richard Cardoso da Silva[1], Divya Singh[1], Marit Leuschner[2], Ina Poser[2], Andrea Ssykor[2], Julian Berlitz[1], Nadine Schmidt[1], Lea Röhder[1], Gerben Vader[1] ✪, Anthony A Hyman[2], Alexander W Bird[1] ✪

**Bacterial artificial chromosome (BAC)–based transgenes have emerged as a powerful tool for controlled and conditional interrogation of protein function in higher eukaryotes. Although homologous recombination-based recombineering methods have streamlined the efficient integration of protein tags onto BAC transgenes, generating precise point mutations has remained less efficient and time-consuming. Here, we present a simplified method for inserting point mutations into BAC transgenes requiring a single recombineering step followed by antibiotic selection. This technique, which we call exogenous/synthetic intronization (ESI) mutagenesis, relies on co-integration of a mutation of interest along with a selectable marker gene, the latter of which is harboured in an artificial intron adjacent to the mutation site. Cell lines generated from ESI-mutated BACs express the transgenes equivalently to the endogenous gene, and all cells efficiently splice out the synthetic intron. Thus, ESI mutagenesis provides a robust and effective single-step method with high precision and high efficiency for mutating BAC transgenes.**

## Introduction

The ability to precisely query functional hypotheses of protein function in cells requires the capacity to express rationally mutated proteins from genes under their native physiological regulation. Traditionally, transgenes expressed in higher eukaryotes are derived from cDNAs, and thus lack native cis-regulatory elements or alternative splicing isoforms, often resulting in overexpression and deregulation artefacts. This may hinder proper phenotypic functional characterization of mutated genes, as well as the determination of the precise localization and interaction partners of the protein products. Whereas the development of sequence-specific

nucleases has enabled mutation of specific genes at their endogenous loci (Gaj et al, 2013; Ran et al, 2013), mutations in essential genes that are lethal will prevent growth and recovery of viable cells. In addition, deleterious mutations are prone to accumulate suppressive changes in chromosome integrity or gene expression during the procedure of selecting and expanding cells for analysis, particularly in the genomically instable cancer cell lines frequently used.

The use of bacterial artificial chromosomes (BACs) as transgenes largely overcomes these limitations. BACs are bacterial vectors containing fragments of eukaryotic genomes that are large enough to encode an entire gene in its genomic context, including cis-regulatory elements. Transgenes expressed from BACs, thus, deliver near-physiological expression in eukaryotic cells (Kittler et al, 2005; Bird & Hyman, 2008; Poser et al, 2008; Bird et al, 2011). Importantly, BAC transgenes modified to be RNAi resistant allow for conditional exposure of recessive mutations through selective depletion of endogenous protein, as an alternative to direct genome editing to overcome the limitations listed above (Bird et al., 2011; Bird and Hyman, 2008; Ding et al., 2009; Kittler et al., 2005; Rondelet et al., 2020; Scolz et al., 2012; Singh et al., 2020; Zheng et al., 2014).

Because of their size, BACs do not lend themselves to restriction/ligation-based modification techniques. Instead, homologous recombination-based recombineering techniques are used to engineer BACs in *Escherichia coli* (Murphy, 1998; Zhang et al, 1998; Yu et al, 2000). Recombineering uses phage proteins (Redα, β, and γ from phage λ; or RecE and RecT from the Rac prophage) to promote homologous recombination, facilitating a wide variety of DNA modifications. The high efficiency of the single-step recombineering procedure to insert a protein tag attached to a selectable marker gene on a DNA vector has allowed its adaptation to genome-scale high-throughput pipelines (Sarov et al, 2006, 2012, 2016; Poser et al, 2008; Hasse et al, 2016). The subsequent generation of transgenic cell lines based on these libraries has further been used to precisely determine the cellular localization and the quantitative interactome of more than 1,000 proteins (Hubner et al,

---

[1]Max Planck Institute of Molecular Physiology, Dortmund, Germany   [2]Max Planck Institute of Molecular Cell Biology and Genetics, Dresden, Germany

Correspondence: alex.bird@mpi-dortmund.mpg.de
Gerben Vader's present address is Section of Oncogenetics, Department of Clinical Genetics, Cancer Center Amsterdam, Amsterdam, The Netherlands
*Arnaud Rondelet and Andrei Pozniakovsky contributed equally to this work

2010; Hutchins et al, 2010; Hein et al, 2015). On the contrary, current techniques to introduce point mutations in BACs still require more intensive work, whether by a counterselection-based two-step procedure (Bird et al, 2011; Wang et al, 2014; Näsvall, 2017; Papa and Shoulders, 2019), a lower efficiency one-step procedure requiring extensive PCR screening (Lyozin et al, 2014), or CRISPR-guided methods (Pyne et al, 2015). Here, we present a simple and efficient one-step procedure to introduce point mutations in BAC transgenes, harnessing introns to carry selectable markers, which reduces the time and cost of generating mutagenized constructs.

Modifying intronic sequences is an attractive approach to modifying or gaining gene functionality with minimal perturbation of the host gene protein product and has met use in a variety of applications such as conditional knockout and mutant constructs, or miRNA expression (Gu et al, 1994; Kaulich et al, 2015; Wassef et al, 2017). Synthetic introns have been designed based on the four core splicing signals necessary for spliceosome processing: two splice sites (SS) located at the 5′ and 3′ intron boundaries (5′SS and 3′SS, respectively), a branch point sequence (BP) located ~25 nucleotides upstream of the 3′SS, and a polypirimidine tract (PPT) directly upstream of the 3′SS (Fig 1A) (Lin et al, 2003; Wang & Burge, 2008; Mercer et al, 2015). Most of the sequence determinants for splicing are thus located within the intron. In most eukaryotes, the consensus sequences of the core splicing elements located within exons are limited to a (C/A) A G sequence directly upstream of the intron and a G downstream of the intron (Fig 1A) (Zhang, 1998). Synthetic introns designed from these minimal core splicing signals are indeed efficiently spliced out of an exon, and have been used in multiple applications, such as making conditional knockouts, or expressing shRNAs or miRNAs (Lin et al, 2003; Greber & Fussenegger, 2007; Lin & Ying, 2013; Seyhan, 2016; Guzzardo et al, 2017).

Here, we present a one-step BAC recombineering procedure in which point mutations are introduced into a BAC along with a synthetic intron. An antibiotic resistance gene located within the intron allows the selection of positive clones after the single recombineering step in bacteria and is seamlessly spliced out of mRNA sequences of the harbouring transgene in eukaryotic cells. This procedure significantly increases the efficiency and decreases the screening time of BAC mutagenesis.

## Results

### Exogenous/synthetic intronization (ESI) mutagenesis: a one-step BAC recombineering strategy for introduction of point mutations

The ESI mutagenesis strategy is outlined in Fig 1. First, a synthetic intron "cassette" is amplified by PCR for use in the recombineering reaction (Fig 1B). This synthetic intron cassette contains the core intron-based splicing signal sequences, as well as an antibiotic resistance gene. We have created multiple cassettes with varying bacterial and eukaryotic selectable markers (Fig S1). To become a functional intron, the synthetic intron cassette must be recombined into a site in the exon where it is directly bordered by an upstream (C/A) A G sequence and a downstream G (Fig 1A and C). Whereas this (C/A) A G | G sequence would occur every 128 nucleotides

assuming random distribution, similar sequences can be mutated to this sequence at amino acid wobble positions during cassette amplification (see, for example, Fig S2), significantly increasing the frequency of useable integration sites. Furthermore, this consensus is not a strict requirement, and a variety of sequences are found in cells and predicted to function (Zhang, 1998; Nguyen et al, 2018; Ohno et al, 2018). Therefore, it is unlikely that a potential synthetic intron integration site will not be found within the useable vicinity of a mutation site of interest. In addition, if the desired mutation is close enough to an existing intron, the antibiotic resistance cassette can be directly targeted to the endogenous intron by generating only one new SS, with the opposite homology arms targeting intronic sequences (see Fig 1C, right side). This is preferable when possible because not only does it maintain more closely the native intron/exon structure but it also avoids the creation of unusually small exons (less than ~50 bp), which may be prone to exon skipping (Dominski & Kole, 1991).

An example of how to design ESI mutagenesis is presented in Fig S2. Once a target site for synthetic intron insertion near the mutation site is identified, primers are designed to amplify the synthetic intron cassette and containing 5′-extended 50-bp arms homologous to sequences directly surrounding the insertion site. One or both homology arms contain the mutation(s) to be integrated. The cassette is amplified by PCR, and the resulting product is inserted into the target BAC via recombineering (Fig 1C). Correct recombination events are selected using the antibiotic resistance encoded within the synthetic intron. A supplementary protocol for the ESI mutagenesis recombineering procedure as well as generating mammalian transgenic cell lines is provided and graphically illustrated in Fig S3 for quick reference.

To compare the efficiency of the ESI mutagenesis technique to a high efficiency, yet more time-consuming method, we generated three different mutations in BACs by either ESI mutagenesis or counterselection (Bird et al, 2011). Table 1 summarizes the result of these analyses. The overall efficiency of the two methods for the given mutations, taking into account all recombineering steps and the final product sequence accuracy, was comparable between the two techniques (37% and 42%). However, ESI mutagenesis is much simpler in design and execution, as well as much faster (~3 versus ~8 d).

Upon transfection of the modified BAC into eukaryotic cells of interest, cells that have integrated the BAC in their genome may also be selected using the antibiotic resistance encoded within the synthetic intron. If desired, the antibiotic resistance gene may be removed by application of Cre recombinase via loxP sites located within the cassette (Fig S1). Once the BAC transgene is transcribed the synthetic intron is spliced out leading to the formation of a messenger RNA carrying the desired mutation.

### ESI-mutated BAC transgenes yield proteins with the expected size and localization

To confirm that BAC transgenes containing synthetic introns yielded proteins of the correct size and localization when expressed in human cells, we started with a panel of 10 GFP-tagged BAC transgenes, and modified each using our ESI mutagenesis strategy to introduce mutations to render them resistant to specific siRNA-targeting sequences (Table S1). We then generated HeLa cell lines

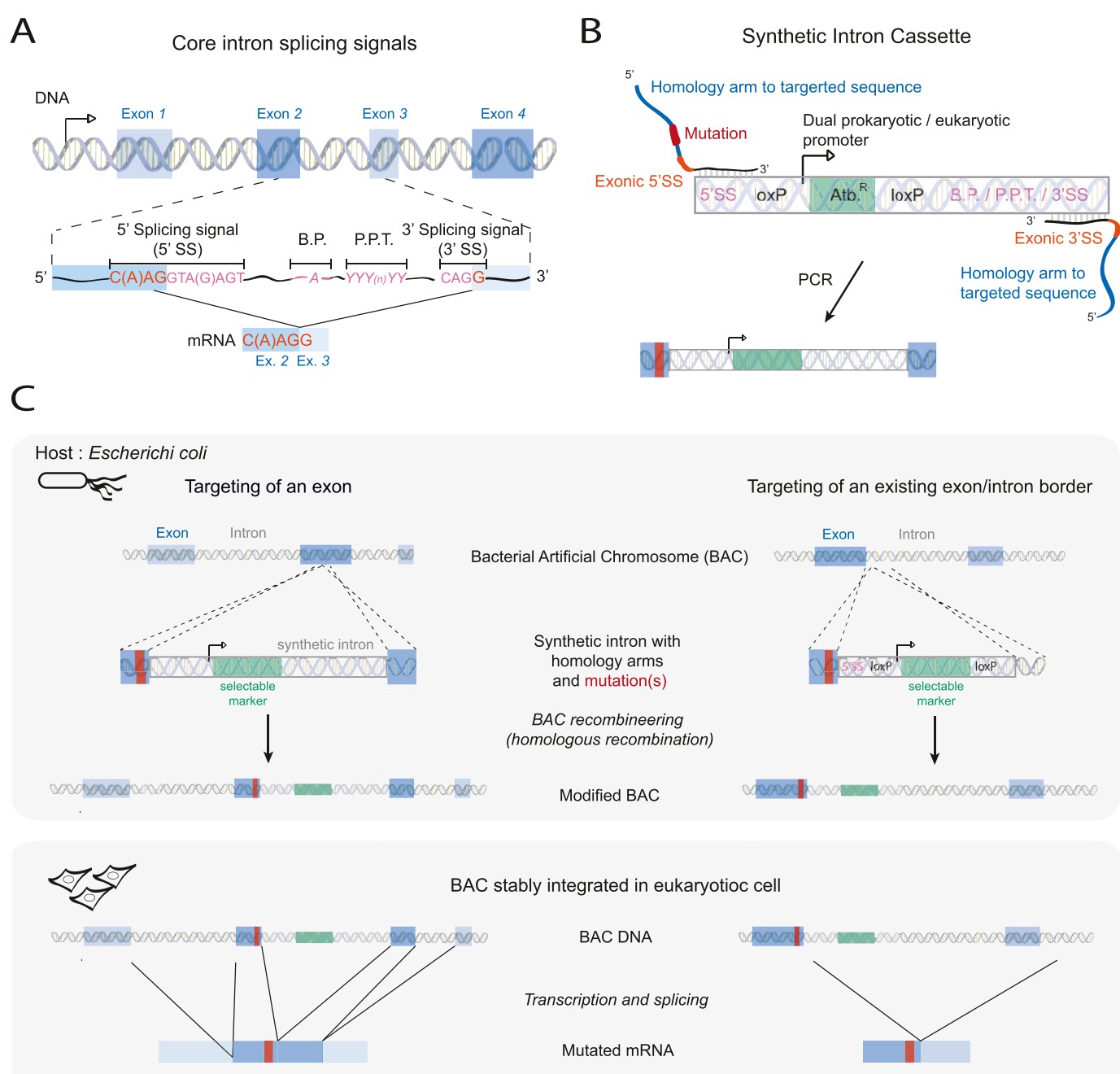

**Figure 1. Exogenous/synthetic intronization (ESI)-mutagenesis: a one-step recombineering procedure to introduce point mutations in bacterial artificial chromosome (BAC).**

**(A)** Scheme showing the organisation of a eukaryotic gene with exons (in blue) and their flanking introns. Introns are spliced out from pre-messenger RNAs to produce mature messenger RNAs (mRNAs) containing only exons. The core splicing signals constitute the minimal information required for the splicing of an intron, and include a 5′ and 3′ splicing signal sequences (5′SS and 3′SS, respectively), a branch point, and a poly-pyrimidine tract (P.P.T). The position of these signals relative to the exon/intron border is indicated with the corresponding nucleotides shown in pink (in introns) or light orange (in exons). ESI mutagenesis is a one-step recombineering procedure that relies on the introduction into a BAC of a synthetic intron coding a selectable marker along with the mutation of interest, thereby allowing for the easy selection of correct recombinants. After transcription of the transgene in eukaryotic cells, the synthetic intron is spliced out to produce a mutated mRNA. **(B)** The synthetic intron constitutes of the intronic core splicing signal (in pink), an antibiotic resistance cassette (Atb.[R]) under the control of a dual eukaryotic/prokaryotic promoter (in green), and two loxP sites flanking the antibiotic resistance cassette. To serve as a template for BAC recombineering, the synthetic intron is amplified by PCR with primers containing ~50-bp-long homology arms to the targeted sequence (in blue), the mutation(s) to introduce (in orange), and the exonic part of the 5′SS and 3′SS (in light orange). **(C)** Scheme showing the introduction of a point mutation into a BAC by ESI mutagenesis. Depending on the position of the sequence to mutate, the point mutation can be inserted along with the synthetic intron into an exon (left panel), or if the mutation is located in the proximity of a pre-existing intron, the part of the synthetic intron encoding the antibiotic resistance can be targeted into this intron (right panel). The synthetic intron is spliced out during RNA maturation, and only the desired mutation is present in the mRNA.

**Table 1.** Comparison of recombineering efficiencies of exogenous/synthetic intronization (ESI) mutagenesis and counterselection.

| Mutations | Artificial intronization (ESI) | | | Counterselection | | | |
|---|---|---|---|---|---|---|---|
| | Insertion (PCR) | Sequencing | Total % | Cassette insertion (PCR) | Rescue (PCR) | Sequencing | Total % |
| GTSE1 (SXIP) | 32/50 | 7/10 | 44.8% | 5/8 | 7/50 | 7/7 | 8.8% |
| GTSE1 (T165A) | 30/50 | 10/12 | 50.0% | 5/8 | 44/50 | 10/10 | 55.0% |
| CHC (C) | 22/100 | 6/7 | 17.1% | 6/8 | 41/50 | 10/10 | 61.5% |
| **Average** | 48% | 80% | 37.3**%** | 67% | 61% | 100% | 41.8**%** |

stably expressing either the "parental" GFP-tagged genes, or the "ESI mutated" (i.e., RNAi resistant) GFP-tagged genes (Fig 2A). Because these mutations should not change the corresponding protein sequence, this allowed us to directly compare the expression and localization of the mutated transgenes to that of the parental, while offering an assay for the presence and function of the introduced mutation (RNAi resistance). Depending on the location of the mutation with respect to an existing intron, we either targeted the synthetic intron along with the desired mutation into an exon (4/10 BACs) (Fig 1C left panel, Fig 2B) or made use of an existing exon-intron border to introduce the desired mutation and the antibiotic resistance (6/10 BACs) (Fig 1C right panel, Fig 2B). All stably integrated "ESI-mutated" transgenes led to the expression of proteins of sizes identical to the one expressed from the corresponding parental transgenes (Fig 2C). Moreover, we could not observe any differences in the cellular localization of proteins expressed from "ESI-mutated" transgenes as compared with those expressed from parental transgenes (Fig 2D). We next performed siRNA experiments on six of the parental/ "ESI-mutated" BAC lines pairs to confirm functionality of the mutations. When the parental BAC-GFP lines were treated with the siRNA against the tagged transgene, we observed a reduction in the corresponding mRNA levels comparable to that observed in HeLa wild-type cells (Fig 3A). In contrast, ESI-mutated BAC lines did not show any reduction in the corresponding mRNA levels upon siRNA treatment. We then analysed RNAi resistance on the protein level for two of the ESI-mutated BAC lines for which we had the corresponding specific antibody. Both CEP135 and AURKB ESI-mutated BAC clonal lines showed expression of their respective GFP-tagged transgenes at a level similar to the corresponding endogenous proteins (Fig 3B), and treatment of both ESI-mutated lines with the corresponding siRNA resulted in depletion of the endogenous protein while the transgene could still be detected (Fig 3B).

To further demonstrate that ESI mutagenesis allows the introduction of functional mutations in a BAC transgene, we used the procedure to mutate two EB1-interacting motifs (SxIP) of the microtubule-associated protein GTSE1 to abolish its interaction with the plus-end tracking protein EB1 (Honnappa et al, 2009; Scolz et al, 2012). In this example, we started with a GFP-tagged GTSE1 BAC, integrated the synthetic intron between the two sites to be mutated and included the two mutations (SxNN) within the 5′ and 3′ homology arms (Fig S2). U2OS cells were generated stably expressing the BAC transgene and analysed for GTSE1 expression and interaction with EB1. The ESI-mutated BAC yielded a protein of the same size as GTSE1 endogenously tagged with GFP, as expected (Fig 3C,

left panel) (Bendre et al, 2016). However, the mutated protein was no longer efficiently pulled-down by GST-EB1 when compared with the endogenous GTSE1 or GTSE1-GFP (Fig 3C), verifying that the mutation was present.

### A functional ESI-mutated BAC transgene is expressed in all cells of a clonal line

We showed above that ESI-mutated BACs yield proteins of the expected size and the correct cellular localization, and that mutation may efficiently confer RNA-resistance or loss of interaction partner when analysed at the cell-population level. We next further assessed whether an ESI-mutated transgene was correctly spliced and functionally expressed, that is, able to sustain the cellular functions of its endogenous counterpart, uniformly in all the cells of a population. This is critical for functional and phenotypic analysis of mutants on the single-cell level. To address this question, we ESI-mutated the gene encoding Aurora A kinase (AURKA) on a GFP-tagged BAC transgene to render it resistant to RNAi, transfected it into U2OS cells, and selected a cell clone (U2OS AURKA-GFP siRES [ESI]) that expressed Aurora A-GFP at the same level as the endogenous Aurora A (Fig 4A, third lane). Treatment of this line with siRNA against AURKA led to the depletion of the endogenous Aurora A, whereas the GFP-tagged Aurora A remained expressed (Fig 4A, fourth lane) and correctly localised to the centrosomes and the spindle (Fig 4B and C). To evaluate the spicing efficiency of the artificial intron, we performed RT-PCR analysis of potential splice products (Fig S4). This analysis showed that Aurora transcripts were correctly spliced (>98%) at the artificial intron.

U2OS cells depleted of Aurora A accumulate in prometaphase and show smaller bipolar spindles (Marumoto et al, 2003; Bird and Hyman, 2008) (Fig 4D–F). In contrast, cells containing the RNAi-resistant Aurora A-GFP ESI maintained a normal distribution of mitotic phases, similar to control-treated cells (Fig 4D). Furthermore, we could not observe any differences in the length of the metaphase spindle between U2OS treated with control-siRNA and the U2OS AURKA-GFP siRES (ESI) clone treated with the siRNA against AURKA (Fig 4E and F). This result indicates that all cells depleted for the endogenous Aurora A express a functional ESI-mutated Aurora A-GFP ESI transgene. Taken together, these data show that all cells of the U2OS AURKA-GFP siRES (ESI) clone uniformly express a functional Aurora A-GFP protein.

To further evaluate the ESI mutagenesis method, we used it interrogate the role of a specific phosphoregulated mitotic protein–protein interaction in chromosome segregation via point mutagenesis. Cdk1

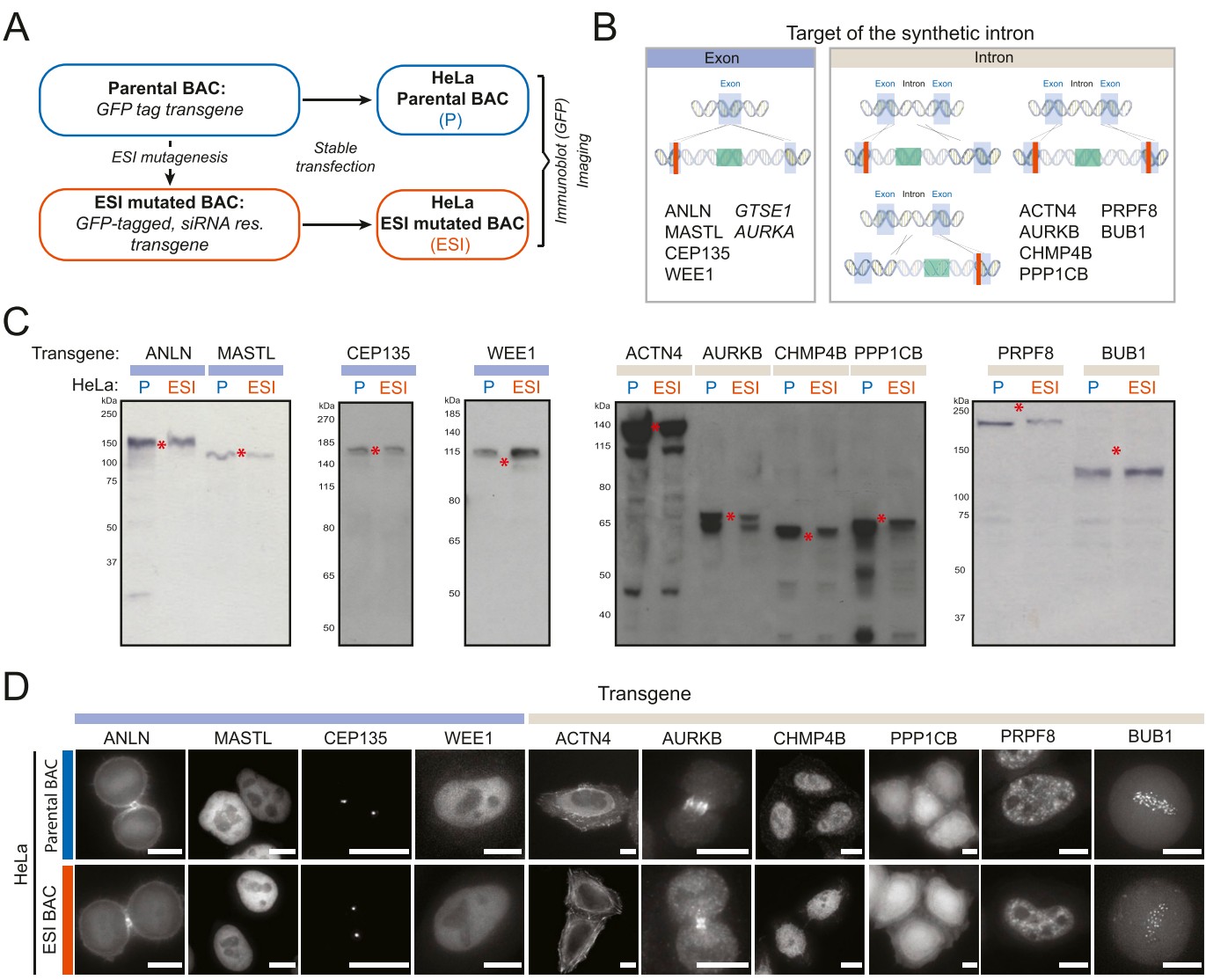

**Figure 2. Exogenous/synthetic intronization (ESI)–mutated bacterial artificial chromosome (BAC) transgenes yield proteins of the right size and localization.**
**(A)** GFP-tagged BAC transgenes (Parental BAC, P) were mutated by ESI mutagenesis to be RNAi resistant (ESI-mutated BAC, ESI) and transfected into HeLa cells. **(B, C, D)** Depending on the position of the sequence to mutate, the synthetic intron was either targeted into an exon (left panel, light blue label in B, C, and D) or into the neighbouring intron by making use of preexisting 5′SS, and/or 3′SS (right panel, light brown label in B, C, and D). Names of the mutated transgene are indicated in the panel corresponding to their mutation strategy. The ESI-mutated *AURKA* transgene is analysed in Fig 4. The *GTSE1* transgene was ESI mutated at its SxIP motifs and is analysed in Fig 3. **(C)** GFP-tagged BAC transgenes ESI mutated to be RNAi resistant yield proteins of the expected size. Immunoblotting on cell lysate of pools of HeLa cells transfected with either parental BACs (P) or ESI-mutated BACs (ESI). GFP antibody was used as a probe. Transgene names are indicated at the top. Red asterisks mark the predicted size of the GFP-tagged protein. **(D)** GFP-tagged transgenes ESI mutated to be RNAi resistant show the same cellular localization as their parental GFP-tagged transgenes. Still images of live cell imaging on HeLa stably transfected with the parental or the ESI-mutated BAC. CHMP4B and AURKB-GFP transgenes were detected by immunofluorescence with anti-GFP antibody. Because of differences in expression level within cell pool, each picture was acquired and scaled independently of the others. Scale Bar 10 μm.
Source data are available for this figure.

kinase phosphorylation of GTSE1 during mitosis disrupts its interaction with the microtubule plus-end tracking protein EB1, and 14 phosphosites coded in exon 9 of GTSE1 are important for this regulation (Singh et al, 2020). We used ESI mutagenesis to mutate these 14 sites to alanines, replacing the intron between *GTSE1* exons 8 and 9 with an artificial intron linked to the mutations, in a BAC previously mutated to confer RNAi resistance (Fig 4G). This mutated BAC was later used to generate a U2OS cell clone stably expressing the transgene (Singh et al., 2020). RT-PCR analysis of splicing efficiency showed that most transcripts were spliced correctly (there was a reduction compared with control that did not reach statistical significance) (Fig S4). Importantly, the protein levels of the GTSE1-14A transgene were not less than the endogenous GTSE1 (Singh et al, 2020). Chromosome segregation efficiency (number of lagging chromosomes) was quantified in this cell line as compared with a wild-type control. As is shown in Fig 4H, the phosphosite mutations led to a significant increase in lagging chromosomes, indicating that failure to disrupt the EB1-GTSE1 interaction during mitosis causes chromosome missegregation defects.

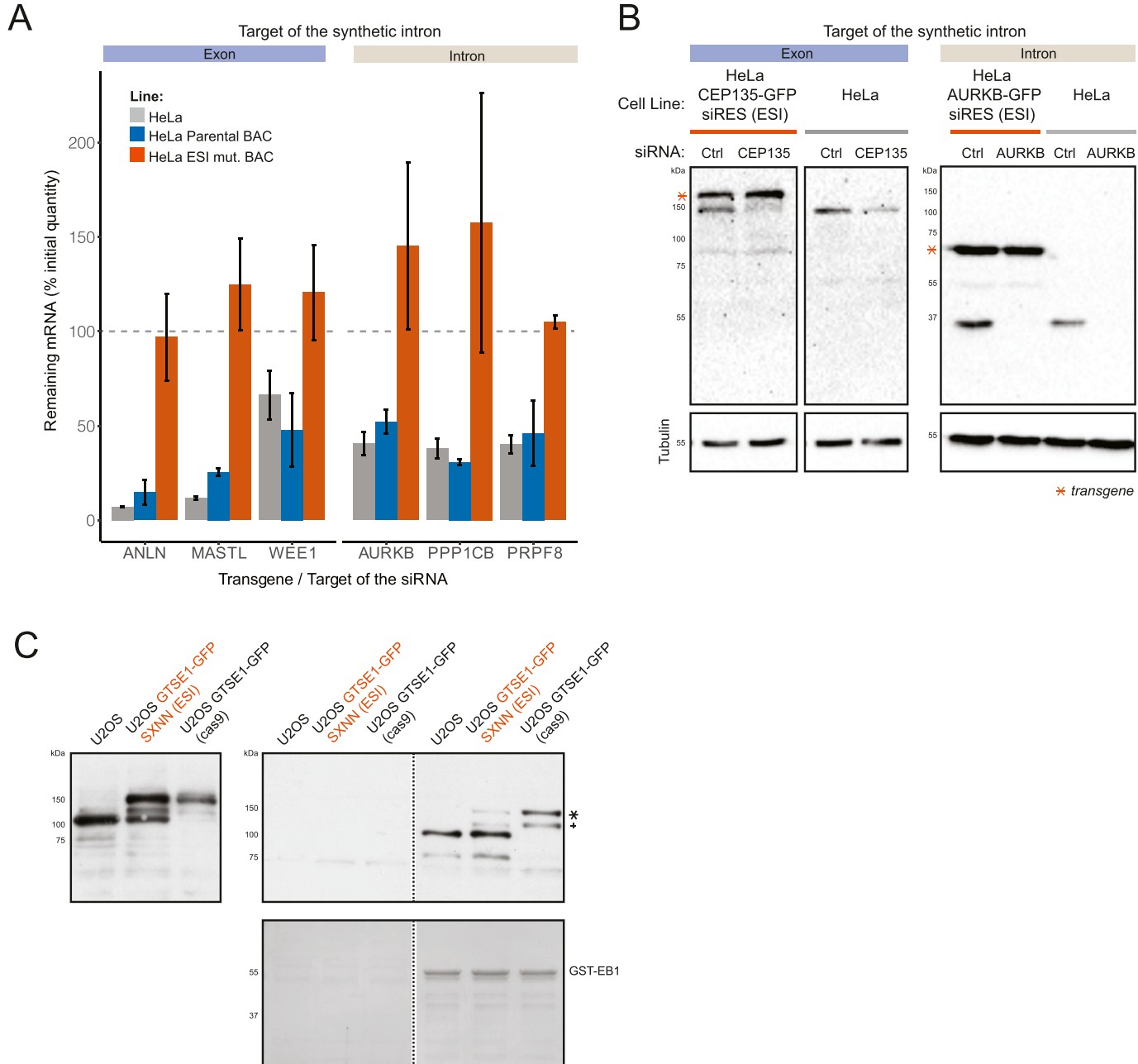

**Figure 3. Exogenous/synthetic intronization (ESI)-mutated bacterial artificial chromosome (BAC) transgenes show the expected phenotype.**
**(A)** BAC transgenes ESI mutated to be RNAi resistant show RNAi resistance at the mRNA level. HeLa wild type, parental BAC lines, and ESI-mutated BAC lines were treated with control-siRNA or transgene-specific siRNAs. Levels of mRNA corresponding to the transgenes were determined by qPCR and normalized to GAPDH mRNA levels. For each cell lines, the % of mRNA remaining in the transgene-siRNA treated cells as compared with the control-siRNA treated cells are presented (N = 2 exp.). A panel of three transgenes mutated by targeting the synthetic intron into an exon (light blue label) and three transgenes mutated by targeting the synthetic intron into a preexisting intron (light brown label) were analysed. Error bars represent SD. **(B)** BAC transgenes ESI mutated to be RNAi resistant show RNAi resistance at the protein level. HeLa and clonal HeLa lines expressing ESI-mutated CEP135 or AURKB-GFP–tagged BAC transgenes were treated with control, CEP135 or AURKB-siRNAs. Levels of AURKB, CEP135, and Tubulin were monitored by immunoblot using specific antibodies. **(C)** Mutation of *GTSE1* SxIP motifs into SxNN by ESI mutagenesis disrupts the interaction of a GTSE1-GFP BAC transgene with EB1. Cell lysates from U2OS, U2OS expressing an ESI-mutated GTSE1-GFP SxNN BAC transgene, and U2OS expressing an endogenously GFP-tagged GTSE1 (GTSE1-GFP Cas9) were used in pull-downs with GST or GST-EB1 as baits. Pull-downs inputs and outputs were probed by immunoblot using GTSE1 antibody. GST and GST-EB1 fusion were visualized by Coomassie Blue.

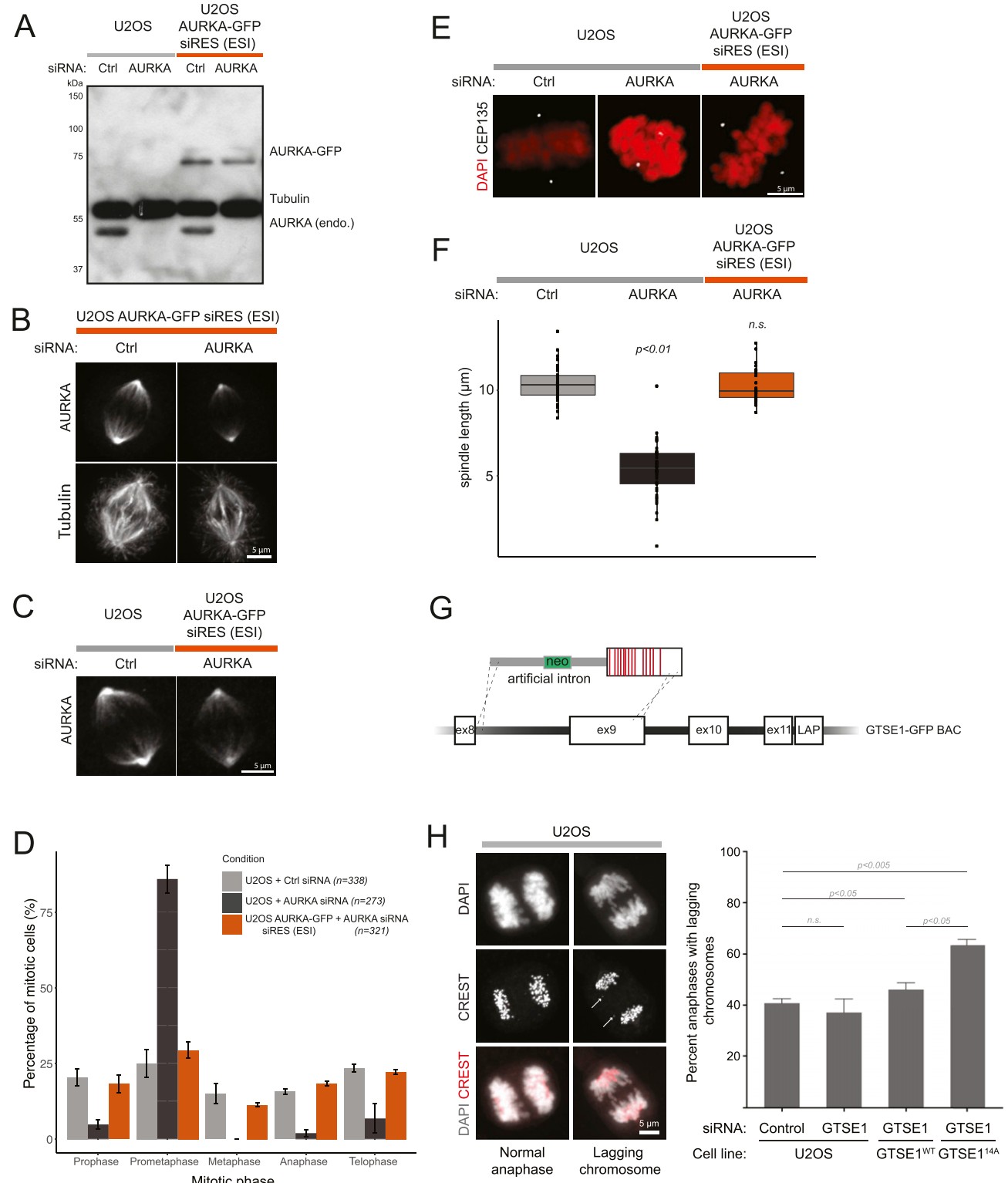

**Figure 4.  An AURKA-GFP bacterial artificial chromosome transgene exogenous/synthetic intronization (ESI)–mutated to carry RNAi resistance rescues endogenous AURKA depletion.**
**(A)** An AURKA-GFP transgene ESI mutated to be RNAi resistant is expressed at endogenous level and is RNAi resistant. Immunoblotting on U2OS and a U2OS clone expressing the ESI-mutated AURKA-GFP bacterial artificial chromosome transgene (U2OS AURKA-GFP siRES [ESI]), both treated with control- or AURKA-siRNA and blocked in mitosis. Antibodies against AURKA and Tubulin were used as probe. **(B, C)** The ESI-mutated AURKA-GFP transgene shows the correct cellular localization. Immunofluorescence on U2OS and U2OS AURKA-GFP siRES (ESI) treated with control- and/or AURKA-siRNA. **(B, C)** AURKA antibody (B) or AURKA and Tubulin antibodies (C)

# Discussion

We have shown that by co-integrating a selectable marker located within an artificial intron, site-directed mutations can be quickly and efficiently introduced into BAC transgenes using a one-step recombineering procedure. This recombineering method is thus analogous to that for integrating protein tags: a single PCR reaction to generate a cassette with homology arms followed by recombineering and selection. The high efficiency of the latter has facilitated high-throughput genome-scale application of protein tagging (Poser et al, 2008; Sarov et al, 2012, 2016; Hein et al, 2015; Hasse et al, 2016). The method we describe here is similar in accuracy to our previously described high-efficient method via counterselection (Bird et al, 2011). However, in contrast, it requires only one recombineering step, easier design principles, and fewer and shorter oligonucleotides, meaning it is both faster and cheaper. Although we have focused on applications in human cell lines, this method would also be applicable to BAC or fosmids commonly used as transgenes in other model organisms.

Critically, BAC transgenes ESI mutated to be RNAi resistant yield proteins of the correct size and localization that can functionally rescue the phenotypes observed upon depletion of the endogenous counterpart. Furthermore, quantitative PCR analyses of splicing products showed that the artificial intron is predominantly correctly spliced. Functional investigation of transgenic lines at the single-cell level supported this finding and further suggested that the synthetic intron is correctly spliced out in all the cells of a population. Furthermore, although we formally describe the ESI mutagenesis technique here for the first time, we have previously successfully used it to generate mutations in BAC transgenes, for example, to functionally analyse a microcephaly associated mutation in the centriolar protein CPAP at the single-cell level in HeLa cells and neural progenitor cells (Zheng et al, 2014; Gabriel et al, 2016).

We cannot exclude that a small portion of synthetic introns within individual cells are mis-spliced, yet without detectable consequences in our functional assays. Indeed, although the core essential sequences for defining introns and SSs are located primarily within the intron and a few nucleotides into the exons, the precise regulation of efficient splicing is a more complex process and impacted by additional sequence information, including that within exons (Chasin, 2007). Nevertheless, mis-spliced mRNAs are probably minimal and degraded by nonsense-mediated decay. It should be noted that as this method is dependent on functional splicing machinery, it may not compatible with investigating effects on splicing pathways, such as spliceosome mutations.

One of our original concerns in this study was that the new synthetic intron or the gene within may impact expression of the transgene to an unuseful extent. Whereas introns have been well characterized to generally increase gene expression (Nott et al, 2003; Shaul, 2017), nested intronic genes found natively in eukaryotic genomes generally impart a negative effect on the "host" gene expression (Yu et al, 2005; Kumar, 2009). Thus, it is possible that expression of a transgene could be impacted by synthetic intron mutagenesis. Although we did not observe gross changes in gene expression in the cell lines generated here, this will likely be highly context-dependent (gene, location of intron, organism, tissue, etc.), and we have observed a case where a nested antibiotic resistance gene in a synthetic intron directly integrated into the genome via CAS9 has decreased host gene expression (unpublished results). We have thus also designed a version of the synthetic intron cassette for BAC recombineering in which the antibiotic resistance only carries a bacterial promoter, eliminating the necessity to recombine it out with Cre. In the end, because BACs integrate randomly into the genome by standard transfection procedures, clonal variations in expression (±2-fold) in transfected cell lines, likely dependent on position effect and copy number, are anyway observed. Such variations could compensate for synthetic intron effects, and it is thus good practice, before phenotypic functional analysis, to select for clonal cell lines expressing BAC transgenes at endogenous levels.

The general strategy outlined here of harbouring a selectable marker gene in an artificial intron adjacent a mutation could in principle be used to increase the efficiency of isolating point mutations via Cas9-induced homologous recombination directly into the genome. This would particularly be useful in hard-to-transfect cell lines where extensive screening may be necessary without a selection step. In our preliminary studies applying this ESI mutagenesis strategy to direct Cas9-mediated human genome modification, we are indeed able to recover modified cell lines at high frequency. We also found, however, more readily observable impacts on gene expression from the synthetic intron when located in the native gene, which could be alleviated by removing the antibiotic resistance gene after Cre application. Although further development of the synthetic intron cassette design/strategy towards direct genome modification holds promise for high efficiency selection of Cas9-mediated site-directed mutation, sensitive applications (i.e., anything to be used in a clinical setting) would not be advisable because of potential changes induced by the synthetic intron.

The use of BAC transgenes remains a powerful genetic tool for exploration of protein function. The ESI mutagenesis method described here allows for straightforward and rapid site-directed mutagenesis of BAC transgenes to further improve their ease of use.

---

were used as a probe. **(D, E, F)** The ESI-mutated AURKA-GFP transgene rescues the mitotic arrest (D) and spindle collapse (E, F) observed after endogenous AURKA depletion. **(D)** The percentage of cells in each mitotic phase is presented (D) for U2OS and U2OS AURKA-GFP siRES (ESI) clone treated with control- or AURKA-siRNA (N = 3 exp; error bars represent standard error). **(E)** Metaphase U2OS and U2OS AURKA-GFP siRES (ESI) cells were treated with control- or AURKA-siRNA and stained by immunofluorescence using DAPI and an antibody against CEP135 (E). **(F)** The pole-to-pole distance ($\mu$m) in metaphase cells is presented (F). In AURKA-depleted U2OS, spindle length was measured in prometaphase cells with nearly aligned chromosomes. The number of mitotic cells used in each condition (n) and statistically significant differences to U2OS treated with control-siRNA are indicated (Kruskal–Wallis, followed by paired Wilcoxon test). **(G)** Schematic showing the construction of the *GTSE1-14A* mutant with an artificial intron. **(H)** Cells containing the GTSE1-14A mutation have a defect in chromosome segregation. Images on the left show normal versus "lagging" chromosomes. Histogram shows the quantification of lagging chromosomes in indicated cell lines/conditions. n ≥ 143 cells from three independent experiments. Error bars indicate SEM. *P*-values from Mann–Whitney test. All scale bars 5 $\mu$m.

# Materials and Methods

## Bacterial artificial chromosomes and recombineering

All BACs used and generated, including modified sequences, are presented in Table S1 except for the *GTSE1-14A* mutant, described in Singh et al, (2020). C-terminal (LAP) and N-terminal (NLAP) GFP-tagged "parental" BACs were generated by recombineering as described in Poser et al (2008). "ESI-mutated" BACs were generated from "parental" BACs following the procedure outlined in Fig S3 and Supplemental Data 1.

## Cell lines and cell culture

U2OS, HeLa cells, and derivatives were grown at 5% $CO_2$ and 37°C in DMEM (PAN Biotech) supplemented with 10% filtered FBS (Gibco), 2 mM L-glutamine (PAN Biotech), 0.1 mg/ml streptomycin, and 100 U/ml penicillin (Pen/Strep mix; PAN Biotech). BACs were transfected into cells using the Effectene kit (QIAGEN) following the manufacturer's instructions, and stable cell lines were selected on antibiotics.

## RNAi

siRNA sequences used are presented in Table S1. siRNAs were purchased from Thermo Fisher Scientific/Ambion. Unless specified otherwise, control-siRNA was Silencer negative control No. 2 (Thermo Fisher Scientific). A reverse transfection approach was used to deplete AURKA and CEP135, and GTSE1 (Bendre et al, 2016), using 50, 40, and 80 nM siRNA, respectively. The transfection procedure for 24-well plates with a final volume of 500 µl is described. If required, the procedure was scaled up. The siRNA and 2.5 µl Oligofectamine (Invitrogen) were prepared separately in a final volume of 50 and 15 µl OptiMEM (Invitrogen), respectively. After 5-min incubation at RT, the siRNA and transfection reagent were mixed and further incubated 20 min at RT. Meanwhile, HeLa cells, U2OS cells, or their derivatives were seeded at the desired confluency into 24-well plates containing or not a coverslip. The transfection mix was finally added onto the cells and the volume adjusted to 500 µl with pre-warmed medium. The medium was changed 7–8 h posttransfection. For CEP135 depletion, the cells were harvested for Western blot 48 h posttransfection. For AURKA depletion, 12-h posttransfection cells were synchronized with 3.3 µM Nocodazole for 18 h and harvested for Western blot. Alternatively, the cells seeded on coverslips were fixed and processed for microscopy 30 h posttransfection. A forward transfection approach was used to deplete AURKB using 50 nM siRNA. The day before transfection, cells were seeded at ~40% confluency in a 3.5 cm dish. The siRNA and 5 µl of Oligofectamine (Invitrogen) were prepared separately into 200 µl OPTIMEM. After 5-min incubation at RT siRNA and Oligofectamine solutions were mixed and further incubated 20 min at RT. Meanwhile, the cells were washed twice with PBS and 1.6 ml OPTIMEM were added on top of them. The transfection mix was then added onto the cells. The cells were harvested for Western blot 48 h posttransfection. In experiments measuring mRNA levels in HeLa cells, a control scrambled RNA (customed; Ambion, sense: 5′-UUCUCCGAACGUGUCACGUtt-3′) was used, and siRNA was transfected using Interferin (Polyplus Transfection) and 100 nM siRNA per well of a six-well plate in forward or reverse transfection (140,000 cells). Duration of the depletion is indicated for each gene in Table S1.

## Antibodies

Mouse anti-GFP (11 814 460 001; Roche), rabbit anti-GTSE1 (custom generated; described in Scolz et al [2012]), and rabbit anti-Aurora kinase B (ab2254; Abcam) were used for Western blots. Mouse anti–α-tubulin (DM1α; Sigma-Aldrich), rabbit anti-Cep135 (custom generated, described in Bird and Hyman [2008]), and goat anti-GFP (MPI Dresden; described in Poser et al [2008]), and goat anti-Aurora A (sc-27883; Santa Cruz) were used for both Western blot and immunofluorescence. Donkey anti-goat Alexa 488 (705 545 147; Jackson Immunoresearch), donkey anti-mouse Alexa 594 (A90-337D4; Bethyl Laboratories), and donkey anti-rabbit Alexa 650 (A120-208D5; Bethyl Laboratories) were used in immunofluorescence. Donkey anti-goat HRP (SC-2020; Santa Cruz), sheep anti-mouse HRP (NXA931-1ml; Amersham), and donkey anti-rabbit HRP (NXA934-1ml; Amersham) were used for Western Blots.

## Western Blot

Cells were lysed on ice for 10–15 min in RIPA buffer or cell lysis buffer (50 mM $Na_2HPO_4$, 150 mM NaCl, 10% glycerol, 1% Triton X-100, 1 mM EGTA, 1.5 mM $MgCl_2$, 50 mM Hepes, pH 7.2, and 1 mM DTE) both supplemented with 2× protease inhibitor mix (SERVA). Debris was eliminated by centrifugation at 4°C at maximum speed in a bench-top centrifuge. Samples were prepared in Laemmli buffer and run on SDS–PAGE before transfer onto a nitrocellulose membrane. 5% milk (Carl Roth) in PBS Tween 20 0.1% (SERVA Electrophoresis) was used to block the membrane before incubation with the primary and secondary antibodies diluted in the same blocking solution. Membranes were incubated with ECL reagent (GE Healthcare) before development onto Hyperfilm (Amersham) or imaging on the ChemiDoc MP imaging system (Bio-Rad). Digital data were obtained by scanning Hyperfilms or using the ImageLab software (Bio-Rad). Fiji was used to adjust levels, generate 8-bit tiffs, and crop images.

## RNA extraction and quantitative RT PCR (RT-qPCR)

For experiments in Fig 3A (mRNA abundance in different cell lines), cDNA was isolated from 250,000 cells per condition using the Cells-to-cDNA II Kit (AM1722; Ambion) following the manufacturer's two-step RT-PCR protocol. qPCR was performed with SYBR Green (Thermo Fisher Scientific) on a Mx3000 qPCR Machine (Stratagene). Primers are indicated in Table S1.

For experiments in Fig S4A (measuring splicing efficiency), total RNA was isolated from U2OS GTSE1[14A] and AURKA-GFP siRES (ESI) with the RNeasy mini kit (QIAGEN) followed by on-column DNase-I digestion, according to the manufacturer's protocol. 1 µg of total RNA was reversed transcribed using the superscript-III reverse transcriptase (Thermo Fisher Scientific) and random primers (New England Biolabs) or anchored Oligo(dT) Primer (Thermo Fisher Scientific), according to the manufacturer's protocol. Reactions were incubated with RNase-H (Thermo Fisher Scientific) to remove RNA complementary to the cDNA. Levels of cDNAs were determined using the absolute quantification by the standard curve method on a

CFX-Connect Real-Time PCR detection system (Bio-Rad). 10-fold serial dilutions absolute standard curves were generated with using purified linearised plasmids as templates. For GTSE1-14A (AI-Exon 9[14A]), GTSE1-14A (Across AI: Exon 8 - Exon 9[14A]), AURKA (AI in Exon 6) and AURKA (Across AI in Exon 6), plasmids (pUCGW) containing each amplicon were ordered from Genewiz (pGV1102-pGV1105). To obtain additional plasmids, the PCR products of AURKA (Control within Exon 5), GTSE1-14A (Control within Exon 10) and GAPDH were blunt-end cloned into the SmaI site of pUC18, generating the plasmids (pGV1106-pGV1108). Linearised plasmids were then gel purified and their concentration fluorometrically determined with PicoGreen reagent (Molecular Probes) using the Quantus fluorometer (Promega). Melting curves were generated for each amplicon to assess the amplification specificity. The estimated number of copies for every amplicon was obtained by plotting the quantification cycle (Cq) values against the linear regression line generated with the serially diluted curve in every reaction. All values were subtracted from the background (reaction devoid of superscript-III) and normalized to *GAPDH*. Oligo sequences, their respective amplification efficiencies, and the Linear regression (amplicon concentration versus *Cq* values) to estimate the number of cDNA copies are available in Table S2. PCR efficiencies (PE) were calculated according to Pfaffl (2001).

### Pull-downs

Cells from a confluent 15-cm were lysed for 15 min on ice in 2 ml cell lysis buffer (50 mM $Na_2HPO_4$, 150 mM NaCl, 10% glycerol, 1% Triton X-100, 1 mM EGTA, 1.5 mM $MgCl_2$, 50 mM Hepes, pH 7.2, and 1 mM DTE) supplemented with 2× protease inhibitor mix (SERVA) and 1 mM DTE (15 min on ice). Debris was removed by centrifugation at maximum speed at 4°C in a bench-top centrifuge. Total protein concentration was measured by Bradford. Equivalent amounts of cell lysate were incubated 20 h at 4°C on GST or GST-EB1 beads. Beads were washed thrice with 1 ml cell lysis buffer supplemented with 2× protease inhibitor. Proteins were eluted with 45 µl Laemmli 2×.

### Immunofluorescence

For most experiments, cells were seeded onto coverslips and fixed in ice-cold methanol for 12 min at −20°C. Cells were blocked at RT with 5% BSA (Sigma-Aldrich) dissolved into PBS. For images of Aurora B-GFP and CHMP4B-GFP cell lines, the cells were fixed in 3% PFA supplemented with 5 mM EGTA, 1 mM $MgCl_2$, and 2% sucrose, they were washed, and then permeabilized using PBS/0.1% Triton X-100 and eventually blocked with PBS/0.2% Fish Skin Gelatin (Sigma-Aldrich). Primary antibodies were diluted into 5% BSA in PBS and incubated onto coverslips for 1 h at 37°C in a wet chamber. The same procedure was applied for secondary antibodies. Coverslips were mounted using Prolong Gold antifade mounting with DAPI (Molecular Probes and Thermo Fisher Scientific). The cells were washed thrice with PBS between every step.

### Microscopy

Immunofluoresence images for the AURKA experiments were acquired using a spinning-disk confocal system (Marianas, Intelligent Imaging Innovation, 3i) built on an Axio Observer Z1 microscope (Zeiss) with an ORCA-Flash 4.0 camera (Hamamatsu Photonics). Images were taken with a 63× 1.4 NA Plan-Apochromat oil objective (ZEISS). The spindle size was measured using the CEP135 signal in the Slidebook software (Marianas, Intelligent Imaging Innovation, 3i, V5.5 or later).

Experiments assessing the localization of the GFP transgenes were all performed on live cells with the exception of the AURKB and CHMP4B transgenes. Live cells were imaged in $CO_2$-independent visualization media (Gibco). Images were acquired using a DeltaVision imaging system (GE Healthcare) with an sCMOS camera (PCO Edge 5.5) and a 60× 1.42NA Plan Apo N UIS2 objective (Olympus).

## Supplementary Information

## Acknowledgements

We thank Mihail Sarov for helpful comments on the manuscript. This work was supported by a Worldwide Cancer Research project grant 16-0093 to AW Bird. The Vader laboratory was financially supported by the European Research Council (ERC Starting Grant URDNA, agreement number 638197, to G Vader) and the Max Planck Society.

### Author Contributions

A Rondelet: conceptualization, formal analysis, investigation, visualization, methodology, project administration, and writing—original draft, review, and editing.
A Pozniakovsky: conceptualization, investigation, and methodology.
D Namboodiri: investigation.
R Cardosa da Silva: formal analysis, investigation, and methodology.
D Singh: investigation and visualization.
M Leuschner: investigation.
I Poser: visualization and project administration.
A Ssykor: investigation.
J Berlitz: investigation.
N Schmidt: investigation.
L Röhder: investigation.
G Vader: resources, supervision, and project administration.
AA Hyman: conceptualization and supervision.
AW Bird: conceptualization, resources, supervision, funding acquisition, methodology, project administration, and writing—original draft, review, and editing.

### Conflict of Interest Statement

The authors declare that they have no conflict of interest.

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
