## [Reviewer comments · Life Science Alliance]

Life Science Alliance

ESI mutagenesis: A one-step method for introducing mutations into bacterial artificial chromosomes

Arnaud Rondelet, Andrei Pozniakovsky, Devika Namboodiri, Richard Cardoso da Silva, Divya Singh, Marit Leuschner, Ina Poser, Andrea Ssykor, Julian Berlitz, Nadine Schmidt, Lea Röhder, Gerben Vader, Anthony Hyman, and Alexander Bird

DOI: <https://doi.org/10.26508/lsa.202000836>

Corresponding authors: Alexander Bird, Max-Planck Institute of Molecular Physiology

Review Timeline:	Submission Date:	2020-07-01
	Editorial Decision:	2020-07-06
	Revision Received:	2020-10-21
	Editorial Decision:	2020-11-02
	Revision Received:	2020-11-23
	Accepted:	2020-11-23

Scientific Editor: Shachi Bhatt

Transaction Report:

July 6, 2020

Re: Life Science Alliance manuscript #LSA-2020-00836

Dr. Alexander W Bird
Max-Planck Institute of Molecular Physiology
Department of Mechanistic Cell Biology
Otto-Hahn-Staße 11
Otto-Hahn-Str. 11
Dortmund 44227
Germany

Dear Dr. Bird,

Thank you for transferring your manuscript entitled "ESI mutagenesis: A one-step method for introducing point mutations into bacterial artificial chromosome transgenes" to Life Science Alliance from Review Commons.

We have now had a chance to read the reviewers' report and your preliminary point-by-point response. Based on the outline you provide, we think that the proposed revisions sound reasonable. As such, we would invite you to submit a revised version of the current study, provided that the issues raised by the reviewers can be convincingly addressed. In light of reviewer #1's comment, we would encourage you to reformat your method described in Figure S3 using a step-by-step protocol format with bullet points and more detailed text, to facilitate the adoption of the methodologies across labs. The current graphic protocol in Figure S3 is rather straightforward and can still be included in the revised manuscript.

In our view these revisions should typically be achievable in around 3 months. However, we are aware that many laboratories cannot function fully during the current COVID-19/SARS-CoV-2 pandemic and therefore encourage you to take the time necessary to revise the manuscript to the extend requested above. We will extend our 'scoping protection policy' to the full revision period required. If you do see another paper with related content published elsewhere, nonetheless contact me immediately so that we can discuss the best way to proceed.

Please note that papers are generally considered through only one revision cycle, so strong support from the referees on the revised version is needed for acceptance.

Thank you for this interesting contribution to Life Science Alliance. We are looking forward to receiving your revised manuscript.

Sincerely,

Reilly Lorenz
Editorial Office Life Science Alliance
Meyerohofstr. 1
69117 Heidelberg, Germany
t +49 6221 8891 414
e contact@life-science-alliance.org
www.life-science-alliance.org

B. MANUSCRIPT ORGANIZATION AND FORMATTING:

*****IMPORTANT:** It is Life Science Alliance policy that if requested, original data images must be made available. Failure to provide original images upon request will result in unavoidable delays in publication. Please ensure that you have access to all original microscopy and blot data images

before submitting your revision.**

We thank the reviewers for their constructive comments

Reviewer #1 (Evidence, reproducibility and clarity (Required)):

****Summary:****

Provide a short summary of the findings and key conclusions (including methodology and model system(s) where appropriate).

*The authors here describe a method to modify bacterial artificial chromosomes (BAC) harbouring gene loci from eukaryotes. When wanting to modify a BAC an antibiotic selection cassette is often included alongside the desired mutation/modification to increase the number of successful recombinants in *E.coli*. Traditionally, this is removed in a second recombination process to leave only the desired modification. The novelty in the procedure described herein is to add a synthetic intron consensus sequence around the selection cassette, which eliminates the need for the subsequent removal of the antibiotic cassette from the BAC before transfection into mammalian cells, saving time and resources. The technique is clever in its simplicity and appears to function for a number of gene loci. The authors validated the correct functioning of the modified BACs for a number of genes using three main assays - transcript level, protein level and localisation.*

****Major comments:****

Are the key conclusions convincing?

The conclusion that the method described generates functional modified BACs is valid.

Should the authors qualify some of their claims as preliminary or speculative, or remove them altogether?

While the method is successfully employed in this study, its efficiency is not quantified in relation to the state-of-the-art as described in the introduction. One assumes it would be more efficient, but this has not been tested empirically in the paper. Does the inclusion of the synthetic intron sequence have an effect on the efficiency of modifying BACs compared to a more typical two-step positive/negative antibiotic selection cassette?

This is a great point brought up by the reviewer, see below for new experiments addressing this.

The functionality of this approach rests entirely on the ability of the target cell to correctly splice out the synthetic intron. The authors are aware of this potential problem as highlighted in the lines below, but do not make efforts to explicitly test splicing. On lines 224-225, the authors state "We cannot exclude that a small portion of synthetic introns within individual cells are misspliced". On lines 230-231 it is stated that "mis-spliced mRNAs are probably minimal and degraded by nonsense-mediated decay". On lines 215-217, the authors describe an "investigation of transgenic lines at the single-cell level" that suggests "the synthetic intron is correctly spliced out in all the cells of the population". How do the authors reach this

conclusion? U2OS and HeLa cells are considered very "robust" and may not show detectable consequences when stressed with an increased level of nonsense-mediated decay. Further, many genes maintain a high level of expression that buffers them against small changes in transcription/splicing. The synthetic intron might have a bigger impact on more tightly regulated genes, so assessing the splicing rate would be essential if the authors wish to advocate their technique as generally applicable.

We have performed RT-PCR assays for splicing which show a minimal effect of the synthetic intron on splicing; see below for experiments/conclusions.

The ability of the synthetic intron to be removed from final transcripts depends on functioning splicing machinery. The authors might emphasise this issue, as spliceosome mutations are important fields of study and might not be compatible with this method.

We have now added text to specifically address this issue in the discussion.

The authors used un-directed integration of each BAC under study. Therefore, it is hard to assess what effect the synthetic intron has, as the authors only ever assess the downstream levels of the correctly spliced, translated and localised protein. The authors themselves state that this can lead to clonal variations in expression of up to 2-fold and on line 250 that this variation "could compensate for synthetic intron effects", but make no effort to test this. Again, lines 267-268 highlight the potential dangers of potential effects of the synthetic introns, but do not test these.

We have now performed RT-PCR assays for splicing which show a minimal effect of the synthetic intron on splicing. We analyzed two different artificial intron-based mutants which were characterized in this study: an RNAi-resistance mutation in Aurora A, and a series of 14 phosphosite mutations in GTSE1. The latter is a new mutant cell line introduced during the revision process (see below). The results of the assays are shown in Figure S4.

Would additional experiments be essential to support the claims of the paper? Request additional experiments only where necessary for the paper as it is, and do not ask authors to open new lines of experimentation.

If not already performed, a large number of bacterial colonies should be screened for the correct modification and frequency of correct ones reported. This frequency - reported for at least three different modifications - would estimate what sort of efficiency this method provides.

The modified region of each BAC should be sequenced and the results reported. The rate of exactly modified clones is important, in case of spontaneous or low fidelity integration of the antibiotic cassette.

The percentage of transcripts that have the synthetic intron correctly spliced out should be measured for some of the BAC constructs used in the study.

A direct head-to-head comparison of this newer method compared to other techniques, or even the authors' own previous two-step approach is necessary to

assess the benefits of this method. Preferably, the experiment would be run in parallel with and without antibiotic selection applied, to show that it drastically improves chances of finding a correct clone.

We have now performed side-by-side recombineering exercises, for three different mutations, with both the artificial-intronization (ESI) technique presented here and the two-step counterselection technique published in Bird et al Nature Methods 2012. The results of these experiments are summarized in Table 1. What they show is that for these mutations, recovery of modified BACs is more efficient for the ESI technique, while sequencing of mutations indicates that the counter-selection technique produces less mutations in the final clones (80% vs 100%). When considering both factors, the final “efficiency”, that is, the chance that picking a single colony from a plate will have a currently modified BAC is approximately the same when averaging the mutations used here (~40% ; Note that efficiencies reported in Bird et al 2010 were higher but only reflected the second “counterselection” step, and were not sequenced for that analysis). This means that growing up and sequencing 5 colonies on average would be more than needed to obtain a perfectly modified BAC.

A major difference in the techniques, however, is that the new ESI technique is much faster, easier, and cheaper. It is a single selection step, and so does not require the extra days/screening, different antibiotic plates and different induction agents, expensive long (120-mer) oligos, and more complicated oligo design principles that the counter-selection method requires. Thus we feel it is a major improvement (indeed it is the preferred method in our lab). We have modified the discussion to highlight these data and points.

Are the suggested experiments realistic in terms of time and resources? It would help if you could add an estimated cost and time investment for substantial experiments.

Repeating the transformation of the BAC and targeting cassette and assessing the recombination efficiency and sequencing should only require existing reagents and take less than a week or two to complete.

Quantitative RT-PCR to assess the percentage of transcripts that have the synthetic intron spliced out would take a little more work. However, this should not be a considerable investment in time or resources for a standard microbiology laboratory and could be completed within a few weeks using modern techniques, such as that described in Londoño et al. 2016.

Repeating all the experiments in parallel would be considerable work and would only be strictly necessary if the authors wish to emphasise the benefits of their method over the many others already in wide use.

Are the data and the methods presented in such a way that they can be reproduced?

Barring the omission of Table S1, which presumably includes exact information on the BACs modified and sequences used etc., there is sufficient other data and methods to allow the experiments to be repeated.

Targeting the ESI procedure to the middle of exons is likely to have a bigger impact for smaller exons as the authors mention on lines 99-100. Making it clear which exon sizes for each gene were successfully targeted in this study would help give some idea of how significant a problem this might be. Perhaps Table S1 contains this information, but it was not provided. It would also help reviewers check the design strategies.

We apologize for inadvertently failing to upload Table S1 on bioRxiv. It has been uploaded now as part of this submission process. This table indeed contains BAC and target sequence information, including the size of the targeted exon (and the 2 “new” resulting exons). Targeted exons range in size from 138bp to 1537bp, and “new” exons are as small as 48bp.

Are the experiments adequately replicated and statistical analysis adequate?

The replication and statistical analysis of the data as presented appear adequate. Figure Legends should state the statistic used to generate error bars.

We have updated figure legends accordingly

Minor comments:

*Specific experimental issues that are easily addressable.
Are the promoters used in the vectors described universally functional? For example, is the PGK promoter functional in yeast?*

The PGK promoter contained in the cassettes is a mammalian promoter, which has also been reported to work in flies.

Are prior studies referenced appropriately?

The manuscript may benefit from the referencing of BAC modification techniques from a wider variety of groups, such as those using CRISPR-guided recombineering (Pyne et al. 2015).

We have added additional citations, including that mentioned above

Are the text and figures clear and accurate?

The body text is very clear save minor typographical or grammatical errors. Regarding figures, some of the coloured text in Figure 1 is somewhat illegible when printed in grayscale.

We have updated the colors in Fig 1

Line 278 - The acronyms LAP and NLAP are not defined/explained.

fixed

Antibody section starting Line 282 may fit better next to Western Blot section.

changed

Figure 2C - The blot images would benefit from arrows to indicate expected sizes of

proteins.

We have added asterisks to mark the predicted sizes

Figure 3A - the graph may benefit from a dashed line at 100% to highlight that values are normalised to controls.

We have added this

Figure 4 - The differences between panels B & C are unclear.

In B, the localization of AurA is compared in the mutagenized line, before or after siRNA of endogenous AurA. In C, the localization of AurA in the mutagenized line is compared to that of a "wildtype" U2OS cell.

Figure 4E - The legend could provide a little more detail on cell cycle stage/status of the captured cells.

We updated the legend to indicate that these are metaphase cells

Do you have suggestions that would help the authors improve the presentation of their data and conclusions?

Lines 23-27 are somewhat unclear and feel out of context. Perhaps the authors could clarify this as a further advantage of using BACs instead of endogenous gene modifications.

We clarified this text.

While not affecting the factual content of the paper, I would advocate that the authors format the method described in Figure S3 into a more detailed text based layout similar to that seen in a typical Nature Methods article. However, this may depend on the format required by any eventual publishing journal.

We have added a detail text-based protocol of the procedure as Supplementary Protocol 1.

That all of the work the paper was carried out in human cell lines and using human genes is a further caveat, but the authors admit this in the discussion and one would assume that most mammalian cells would respond similarly in their ability to splice out the synthetic intron.

Reviewer #1 (Significance (Required)):

Describe the nature and significance of the advance (e.g. conceptual, technical, clinical) for the field.

This work is a formal description of a newer method that could be useful for many of those employing bacterial artificial chromosomes in numerous studies, such as gene regulation.

Place the work in the context of the existing literature (provide references, where appropriate).

This work builds on methodology previously published by the authors - a counter-selection two-step procedure (Bird et al. 2011). It sets out to formally describe a method merely mentioned as "BAC intronization" in a later paper by some of the authors (Zheng et al. 2014). Other alternative one-step procedures are also available, but present a different set of challenges (Lyozin et al. 2014). Some newer approaches, such as those using CRISPR-guided recombineering (Pyne et al. 2015) or systems that combine CRISPR and positive/negative selection cassettes (Wang et al. 2016) may be slightly more efficient, but are also more complex in their design.

Bird et al. 2011 DOI: 10/dv776q

Pyne et al. 2015 DOI: 10/f7jx92

Wang et al. 2016 DOI: 10/f89db5

Zheng et al. 2014 DOI: 10/f5pkr6

State what audience might be interested in and influenced by the reported findings.

As a technology paper this work should have interest from a broad field of research. While the use of BACs could sometimes be considered more traditional in light of the explosion in CRISPR-based genome editing capabilities, it is definitely seeing a resurgence as the limitations of CRISPR in modifying large regions of genome become more apparent. Therefore, technologies that accelerate the modification of BACs could prove increasingly useful. As category of audience, all those involved in significant recombineering or gene/genome engineering would potentially benefit.

Define your field of expertise with a few keywords to help the authors contextualize your point of view. Indicate if there are any parts of the paper that you do not have sufficient expertise to evaluate.

Synthetic genomics, synthetic biology, cancer cell biology, gene and genome engineering

REFEREES CROSS COMMENTING

I would agree with reviewer two's assessment that we both view the paper in a similar light.

Reviewer #2 (Evidence, reproducibility and clarity (Required)):

This is a methods-focused paper that presents a strategy to efficiently introduce mutations into a bacterial artificial transgene using synthetic introns. BAC-based methods have been an effective strategy for introducing trans genes into human cells to achieve near-endogenous expression, including extensive work from these authors. However, generating mutations and changes within the internal coding sequence presents some challenges for how to target these mutations and select for the mutated form. Here, the authors describe a way to overcome this by introducing

synthetic introns into an adjacent sequence. This allows them to introduce a selectable marker and conduct the molecular biology without creating complications downstream for the functionality of the protein.

This method is carefully described and presented. The authors also provide clear validation by using this to create RNAi-resistant versions of multiple different mitotic factors as well as creating targeted mutants that alter the functional properties of a protein. This work clearly takes advantage of other ongoing studies from these labs (including mutants and cell lines that appear to also have been described elsewhere), but the ability to combine these in a single paper and clearly describe the method provides a helpful advance and validation.

Based on the description and data presented, I think that things are clear and carefully validated. As such, I do not have technical comments or concerns and I would be comfortable with this paper appearing in an appropriate journal in its present form.

Reviewer #2 (Significance (Required)):

This is a solid methods paper, but for considering the nature of the impact and significance of this paper, there are several things to note:

1. The BAC-based method does appear to be a powerful and effective strategy. However, beyond the work of Mitocheck and the authors that are part of this paper, this has not seen widespread adoption. It is possible that this current method may increase its usage due to the value of the targeted mutations within the coding sequence, but at present it is not a broadly used strategy.

We agree that using BACs as transgenes has not seen widespread adoption as a tool in the broader cell biology community (although certainly far beyond members of the Mitocheck consortium). This is likely because many erroneously think that it is a technique for specialist laboratories. We are trying to change this! For reasons outlined below, there is still an increasing desire for conditional analysis of mutated genes under physiological expression/regulation frequently not attainable via directed Cas9-based mutation. A major aim of this paper is thus to further simplify the methods for generating modified BAC transgenes.

2. This BAC-based approach (and also RNAi) are becoming increasingly replaced by the use of CRISPR/Cas9 genome editing. The absence of Cas9-based strategies in this paper limits the potential impact and reach of this paper. The authors do mention the possibility of using a similar synthetic intron strategy for use with Cas9 in the Discussion, and appear to have conducted some experiments. If possible, it would substantially increase the value of this paper if this data and strategy were also included in the Results section (acknowledging that this may still be a work in progress).

While some uses of BAC transgenes are in some cases better replaced by CRISPR/Cas9 techniques (i.e. GFP tagging), there are several occasions where using BACs are preferable: As stated in the text, RNAi-resistant BACs allow for conditional analysis of recessive mutations. Mutations in essential genes that are lethal will prevent growth and recovery of viable cells if integrated into the genome via

Cas9. Additionally, deleterious mutations are prone to accumulate suppressive changes in chromosome integrity or gene expression during the procedure of selecting and expanding Cas9-modified cells for analysis, particularly in the genomically instable cancer cell lines frequently employed. We use both BACs and CRISPR/Cas9 in our lab according to our needs.

We do have an ongoing project to apply this intronization technique to enable more efficient selection of CRISPR/Cas9 integrations. Preliminary results suggest that it works to allow selection of point mutations, but it is still being optimized, including a redesign of the cassette, and is not ready for publication.

3. The method is solid and well-validated, but there are no new results or insights presented in this paper from the work that is described (this is fine, just commenting for considering the right journal fit).

We agree, in the first version as evidence for “biological insights” gained as a result of this technique, we had cited a couple studies that made use of the technique already to functionally analyze a microcephaly-associated mutation in the centriolar protein CPAP at the single cell level in HeLa cells and neural progenitor cells (Zheng et al 2014, Gabirel et al 2016).

We have now added new unpublished data investigating a specific question: Is the cell-cycle-regulated disruption of the EB1-GTSE1 (microtubule plus-end tracking proteins) interaction in mitosis required for chromosome segregation fidelity? We have generated a GTSE1 mutant with 14 phosphosites mutated to alanine using this technique. This experiment revealed that disrupting the phosphoregulation of this interaction compromises chromosome segregation. This new data is now presented in Figure 4.

REFEREES CROSS COMMENTING

It appears that both reviewers are largely on the same page regarding this paper.

November 2, 2020

RE: Life Science Alliance Manuscript #LSA-2020-00836R

Dr. Alexander W Bird
Max-Planck Institute of Molecular Physiology
Department of Mechanistic Cell Biology
Otto-Hahn-Staße 11
Otto-Hahn-Str. 11
Dortmund 44227
Germany

Dear Dr. Bird,

Thank you for submitting your revised manuscript entitled "ESI mutagenesis: A one-step method for introducing mutations into bacterial artificial chromosomes". We would be happy to publish your paper in Life Science Alliance pending final revisions as requested by the reviewers, and necessary to meet our formatting guidelines.

Along with the points listed below, please also attend to the following.

- please upload both your main and supplementary figures as single files
- please upload your main manuscript text as an editable doc format
- please use the [10 author names, et al.] format in your references (i.e. limit the author names to the first 10)
- please add a legend for your Table S1 to the main manuscript text
- please add a callout for your Table 1 and Table S2 to the main manuscript text
- please add the text from the "Supplementary Protocol" file to the main manuscript, if possible. Supplementary documents are usually not downloaded when the readers access the pdfs, and so we encourage you to include all data in the main manuscript file
- please also provide the source data (original unedited images) for Figure 2C

A. FINAL FILES:

B. MANUSCRIPT ORGANIZATION AND FORMATTING:

Sincerely,

Shachi Bhatt, Ph.D.
Executive Editor
Life Science Alliance
<https://www.lsjournal.org/>

Reviewer #1 (Comments to the Authors (Required)):

This strategy and method provides a valuable contribution to the literature. This paper is substantive and interesting, and ideally this approach will be useful and adopted by a range of laboratories. For this version, the authors have addressed each of my comments, and I fully support its publication in LSA. Congratulations on the nice work.

Reviewer #2 (Comments to the Authors (Required)):

The authors here describe a method to modify bacterial artificial chromosomes (BAC) harbouring gene loci from eukaryotes. When wanting to modify a BAC an antibiotic selection cassette is often included alongside the desired mutation/modification to increase the number of successful recombinants in E.coli. Traditionally, this is removed in a second recombination process to leave only the desired modification. The novelty in the procedure described herein is to add a synthetic intron consensus sequence around the selection cassette, which eliminates the need for the subsequent removal of the antibiotic cassette from the BAC before transfection into mammalian cells, saving time and resources. The technique is clever in its simplicity and appears to function for a number of gene loci. The authors validated the correct functioning of the modified BACs for a number of genes using four main assays - transcript level together with protein level, localisation and function.

I believe that the authors have made a commendable effort to address all the points raised by both reviewers. The paper and the data within are of sufficient quality to warrant publication. In particular, their decision to include a detailed protocol as supplemental to the paper along with Figure S2 illustrating an actual example of an ESI design will be incredibly useful for groups wanting to try this technique. Since initial submission, the authors have gone further and used the method to address a biological question. This new data demonstrating the importance of the phospho-regulated EB1-GTSE1 interaction in chromosome segregation is strong evidence that ESI as a technique can help provide new biological insight. To simplify access to this technique, I would encourage the authors to deposit their plasmid templates to a repository, such as Addgene.

One minor issue - Figure 1C has Lorem ipsum text.

November 23, 2020

RE: Life Science Alliance Manuscript #LSA-2020-00836RR

Dr. Alexander W Bird
Max-Planck Institute of Molecular Physiology
Department of Mechanistic Cell Biology
Otto-Hahn-Staße 11
Otto-Hahn-Str. 11
Dortmund 44227
Germany

Dear Dr. Bird,

Thank you for submitting your Research Article entitled "ESI mutagenesis: A one-step method for introducing mutations into bacterial artificial chromosomes". It is a pleasure to let you know that your manuscript is now accepted for publication in Life Science Alliance. Congratulations on this interesting work.

*****IMPORTANT:** If you will be unreachable at any time, please provide us with the email address of an alternate author. Failure to respond to routine queries may lead to unavoidable delays in publication.*******

DISTRIBUTION OF MATERIALS:

Again, congratulations on a very nice paper. I hope you found the review process to be constructive and are pleased with how the manuscript was handled editorially. We look forward to future exciting

submissions from your lab.

Sincerely,

Shachi Bhatt, Ph.D.

Executive Editor

Life Science Alliance

<https://www.lsjournal.org/>
